# Estimation of Early Postmortem Interval from Long Noncoding RNA Gene Expression in the Incised Cutaneous Wound: An Experimental Study

**DOI:** 10.3390/biomedicines10112919

**Published:** 2022-11-14

**Authors:** Mona M. Ali, Samah F. Ibrahim, Noha M. Elrewieny, Aya M. Elyamany, Wagdy K. B. Khalil, Aziza B. Shalby, Sarah A. Khater

**Affiliations:** 1Forensic Medicine and Clinical Toxicology Department, College of Medicine, Cairo University, Cairo 11956, Egypt; 2Forensic Medicine and Clinical Toxicology Department, College of Medicine, Taif University, Taif 21944, Saudi Arabia; 3Clinical Sciences Department, College of Medicine, Princess Nourah bint Abdulrahman University, P.O. Box 84428, Riyadh 11671, Saudi Arabia; 4Pathology Department, College of Medicine, Cairo University, Cairo 11956, Egypt; 5Cell Biology Department, Biotechnology Research Institute, National Research Center, Cairo P.O. Box 12622, Egypt; 6Hormones Department, National Research Center, Cairo P.O. Box 12622, Egypt; 7Forensic Medicine and Clinical Toxicology Department, Faculty of Medicine, Misr University for Science and Technology, 6th October, P.O. Box 77, Cairo 3236101, Egypt

**Keywords:** forensic medicine, incised wound, postmortem interval, long non-coding RNA, matrix metalloproteinase-9

## Abstract

The assessment of alteration of postmortem RNA expression has forensic significance in estimating postmortem interval. To evaluate wound healing progression and the effect of different postmortem intervals, histopathological changes, immunohistochemical matrix metalloproteinase-9 (MMP-9) expression, and long noncoding fatty acid oxidation (lncFAO), RNA expression was assessed in the incised cutaneous wound model. A full-thickness cutaneous wound was inflicted on 75 rats. All 15 rats were sacrificed at different post-infliction intervals (0, 2, 4, 8 and 10 days), and the cutaneous wounds (*n* = 5) were excised at different postmortem intervals (0, 5, and 24 h after euthanasia). The maximal inflammatory healing stage was detected at day 4 post-infliction, while near complete healing, thick mature collagen deposition was detected at day 10 post-infliction. LncFAO expression was significantly over-expressed with increasing wound age. MMP-9 was detectable on injury day with continuous elevation until 8 days post-wounding, which later decreased. Although histopathological and immunohistochemical examinations within 24 h postmortem did not show any remarkable changes, lncFAO RNA expression showed a significant negative correlation with hours passed since death. The combined use of histopathological changes, immunohistochemical expression of MMP-9, and molecular expression of lncFAO could be appropriate in wound dating verification. Among these factors, lncFAO could be a reliable indicator in postmortem interval estimation.

## 1. Introduction

Postmortem interval estimation has a significant impact in a medicolegal context. Determination of vitality and wound dating is the primary goal of a forensic pathologist in trauma deaths, which can be estimated from its healing process involving inflammatory, proliferative, and remodeling stages [1].

During wound healing, inflammatory cellular aggregation, inflammatory marker secretion, angiogenesis, re-epithelialization, and collagen deposition are modulated by various local and systemic factors that can be identified by morphological, immuno-histochemical, and bio-molecular methods [2]. 

These methods analyze inflammatory cellular aggregation (e.g., neutrophilic granulocytes, lymphocytes, macrophages, and fibroblasts) [3], inflammatory substance secretion (e.g., cytokines, growth factors, signaling factors and matrix metalloproteases (MMPs) [4], and long noncoding RNAs (lncRNAs) expression (e.g., lncRNA8975-1, lncRNARP11-586K2.1, lncMALAT1, and lncFAO) [5,6,7].

MMPs, zinc-containing extracellular matrix degradation inhibitors, are subdivided into collagenases, gelatinases, stromelysins, and matrilysins, according to their membrane nature specificity [4,8].

MMPs are necessary at each stage of wound healing due to their role in extracellular matrix reconstruction; however, the complete role of only a few MMPs has been identified [9]. For example, collagenases, including MMP-1, MMP-8, and MMP-13 are involved in wound re-epithelialization; gelatinases, including MMP-2, and MMP-9 are essential in angiogenesis and wound remodeling, while stromelysins, including MMP-3 and MMP-10, are vital for normal cellular proliferation and wound contraction [10].

MMP-9 is a gelatin binding MMP that is highly expressed in many tissues, including skin during pro-inflammatory states. It has many substrates, including gelatins, collagens, and fibrine. During wound healing, it can inhibit cellular proliferation, enhance angiogenesis through activation of proangiogenic cytokines, and promote adipocyte maturation [11]. 

Long noncoding RNAs are noncoding RNA segments with more than 200 nucleotides that can regulate variable biological processes, including wound healing. They regulate cell behavior during wound healing process by acting as signals, decoys, guides, and scaffolds [2].

lncRNAs have a significant function in the wound healing process, such as MALAT1, which is involved in fibroblast migration and endothelial proliferation [12,13]; H19, which promotes cutaneous re-epithelization [14]; and GAS5 and FAO, which regulate macrophages’ transition [5,15].

The gene expression of long noncoding fatty acid oxidation (LncFAO) is up-regulated by injury-induced inflammation to promote inflammation resolution and tissue repair by suppressing inflammatory cytokines, including interleukins, e.g., IL1 and IL6, and tissue growth factor-β, and by suppressing proinflammatory activated macrophages [5]. 

The expressed lncRNAs can be detected earlier than the released inflammatory substance and the changed histological features after injury infliction. Thus, MMPs-based assays and histological features detection are more prominent during late healing stages [1,5].

In the medicolegal field, a combination of morphological and molecular approaches are needed to achieve more precise and reliable determination of wound vitality and dating [16].

The time passed since death, termed as postmortem interval (PMI), has physical and physiological impacts on the wound healing process (e.g., RNA degradation, and microscopical cellular alteration) [17,18]. Although there has been significant progress in the molecular understanding of wound healing, the exact biomolecular effects of different postmortem intervals remain unknown. This study aims to assess histopathological features and expressions of lncFAO RNA and MMP-9 involved in cutaneous incised wound healing during different early PMIs.

## 2. Material and Methods

### 2.1. Ethical Approval

The Institutional Animal Care and Use Committee at Cairo University approved the protocol of this experimental study with reference number (CU/III/F/52/21), Cairo, Egypt.

### 2.2. Animal

The total number of the animals/group was calculated using the resource equation method [10], with an acceptable range of the inter-subject error (DF) 10–20 and number of groups (k) 15. The minimal calculated number of the rats/group was three.
$$\mathrm{Number}\;\mathrm{of}\;\mathrm{animals}\;\mathrm{per}\;\mathrm{group}=\frac{\mathrm{Inter}-\mathrm{subject}\;\mathrm{error}\;\left(\mathrm{DF}\right)}{\mathrm{Number}\;\mathrm{of}\;\mathrm{groups}\;\left(K\right)}+1$$

A total of 75 eight-week-old Wister albino male rats with body weight (200 ± 30 g) were purchased from the animal house of the Faculty of Medicine, Cairo university, Egypt. Each three rats were housed in one plastic cage in experimental conditions (normal day light, temperature of 25 ± 5 °C, and humidity of 45–55%). During experimental duration, balanced diet ad libitum, with clean water was introduced to the them. 

### 2.3. Excisional Wound Model and Sampling

After 7 days of acclimatization, rats were anaesthetized by injecting ketamine and xylazine [11]; hair on the dorsal right side of the rats was shaved and 0.5 cm full thickness cutaneous circular wound was inflicted using punch biopsy technique. The injured skin was left untreated for 10 days. 

Rats were injured at same time and sacrificed (*n* = 15) at different post-infliction intervals (0, 2, 4, 8 and 10 days after wound infliction). To assess the early postmortem interval effect in each post-infliction interval, the cutaneous wounds (*n* = 5) were excised at different postmortem intervals (0, 5, and 24 h after euthanasia). 

Injured cutaneous samples were divided into two parts. Part was used for lncFAO RNA expression analysis and was kept in liquid nitrogen, while the other part was used for MMP-9 expression, histopathological examination, and kept in 10% formalin.

### 2.4. Histopathological Examination

Neutral buffered formalin with 10% concentration was used in full thickness wounded skin specimen’s fixation. After processing and paraffin imbedding, these specimens were sectioned at 4 μm thickness, mounted on glass slides, and stained by hematoxylin and eosin, Masson’s Trichrome stains for microscopic evaluation according to Khalaf et al. [1] and Abu-Albasal et al. [12].

Congestion, edema, hemorrhage, inflammatory cell infiltration, fibroblast proliferation, re-epithelialization, angiogenesis, and collagen deposition were used in wound tissue sections grading. Each parameter was evaluated at 40× magnification and was categorized from 0 to 3 that ranged from no to marked alterations, respectively.

### 2.5. Detection of Matrix Metalloproteases-9 (MMP-9) Expression

Frozen sections were sliced at a 4 μm thickness and mounted on charged slides. MMP-9 immunostaining was done using fully automated immunohistochemical system, Dako autostainer, (pH 6.0) for 10 min. All sections were incubated with rabbit polyclonal MMP-9 antibody (cat. no. ab38898; Abcam, Cambridge, MA, USA) at a dilution of 1:250 at 4 °C overnight. MMP-9 positive staining was identified as brown particles within cellular cytoplasm. It was scored according to Niedecker et al., 2021 [8] from 0 to 3 that expressed no staining (score 0), positive staining of single cells (score 1), positive staining of cell groups (score 2), and positive staining of large tissue areas (score 3), respectively. 

### 2.6. Detection of Long Noncoding Fatty Acid Oxidation (lncFAO) Gene Expression

#### 2.6.1. Isolation of Total RNA 

Total RNA was isolated from homogenized cutaneous tissues (50 mg) by the standard TRIzol^®^ Reagent extraction method (cat#15596-026, Invitrogen, Düren, Germany). Total RNA was treated with 1 U of RQ1 RNAse-free DNAse (Invitrogen, Germany) to digest DNA residues. Purity of total RNA was assessed spectrophotometrically at 260/280 nm ratio, and it was between 1.8 and 2.1. Reverse transcription was done immediately otherwise purified RNA was stored at −80 °C.

#### 2.6.2. Reverse Transcription Reaction

The isolated complete Poly(A)^+^ RNA was reverse transcribed into cDNA in a total volume of 20 µL using RevertAid^TM^ First Strand cDNA Synthesis Kit (MBI Fermentas, St. Leon-Rot, Germany) according to manufacturer’s instructions.

#### 2.6.3. Real Time-Polymerase Chain Reaction

Copy number of rats’ cutaneous samples were identified using StepOne™ Real-Time PCR System from Applied Biosystems (Thermo Fisher Scientific, Waltham, MA, USA).

The sequences of the lncFAO gene primer were as follow; forward: TGC TAC CTC GGT GCT AC, and reverse: TGT TGC TAG GCA CTG GAA AA. The results were normalized to β-actin (forward: AGG GTG TGA TGG GTA TG, and reverse: TCA TCT TTT CAC GGT TGG CC) [4].

Reaction mixtures (25 μL) consisted of 6.5 μL distilled water, 0.5 μL 0.2 μM sense primer, 5 μL of cDNA template, 0.5 μL 0.2 μM antisense primer, and 12.5 μL 1× SYBR^®^ Premix Ex TaqTM (TaKaRa, Biotech. Co. Ltd., Dalian, China) and were used to setup PCR reactions. 

Three steps were allocated as a reaction program. The first step was kept at 95.0 °C for 3 min; the three cycles second step (repeated 40 times) were kept at 95.0 °C for 15 s, at 55.0 °C for 30 s, and at 72.0 °C for 30 s and the final third step (repeated 71 times) was started at 60.0 °C and then increased approximately 0.5 °C every 10 s up to 95.0 °C. The quality of the used primers was checked by the melting curve analysis at 95.0 °C. The relative quantification of the lncFAO to the β-actin was determined by using the 2^−ΔΔCT^ method [13]. 

## 3. Statistical Analysis

Statistical package SPSS version 24 was used to analyze the coded data using One-Way Analysis of Variance (ANOVA) with multiple comparisons post hoc test to compare between groups. Mean and standard deviation (SD) was used to present quantitative variables. All data from the injured cutaneous samples at different PMIs were compared with the control group at 0 h postmortem. *p*-values ≤ 0.05 were considered significant. 

## 4. Results

### 4.1. Morphological and Histopathological Findings

The morphological assessment of the wounds revealed that the wound margins and bed on infliction day were hyperemic. Over 10-day timespan, the wound margins and beds became more contracted and smaller in size to reach near complete healing at the tenth day. No obvious changes were observed in the early post-mortem interval (within 24 h), but inflammatory signs including hyperemia became less prominent on the tenth day (Figure 1).

With the prolongation of post-wounding time, the histopathological inflammatory reaction was gradually increased and plateaued up at day 4, where the ulcerated skin revealed intense dermal inflammatory reaction with congestion, polymorph nuclear cells infiltration, vascular formation, hemorrhage, and edema (Figure 2). 

The semi-quantitative score of the inflammatory parameters was presented in (Table 1).

The wound repair and remodeling increased on day 4 and reached the maximal enhancement on day 10, where the injured skin showed a high epidermal epithelization and a thick mature collagen formation with disappearance of cellular infiltration. With prolongation of PMI up to 24 h, the injured samples did not exhibit any notable histopathological autolytic changes, including nuclear pyknosis, fragmentation, or lysis (Figure 2 and Figure 3).

### 4.2. Immunohistochemical Study of Matrix Metalloproteases-9 (MMP-9) Expression

During wound healing, MMP-9 was expressed at high levels, reaching score 3 at 8 days after wounding, before returning to its lower levels 10 days after wounding. Post-mortem MMP-9 expression in injured samples was not distinguished from ante-mortem expressions during early post-mortem duration of 24 h (Figure 3).

### 4.3. Long Noncoding Fatty Acid Oxidation (lncFAO) Gene Expression

The expression levels of lncFAO gene in injured cutaneous tissues at the same day of wound infliction (control samples) were relatively similar at different postmortem intervals (0.9 ± 0.2, 0.95 ± 0.3, and 0.93 ± 0.3, respectively) (Table 2 and Figure 4). 

However, after 2, 4, 8, and 10 days of wound infliction, the expression levels of lncFAO gene were significantly over-expressed in skin tissues collected immediately after euthanasia (0 h postmortem) compared to the control samples (1.95 ± 0.7, 2.8 ± 0.8, 5.45 ± 0.5, and 10 ± 2.5, respectively) (Table 2 and Figure 4). 

At the 2nd and the 4th day post-infliction, the expression levels of lncFAO gene were progressively decreased with increasing postmortem interval to reach the significant (*p* < 0.05) lowest level at 24 h postmortem (1.6 ± 0.5 and 2.2 ± 0.2, respectively). Additionally, at 8th and 10th day post-infliction, the expression levels of lncFAO gene were significantly (*p* < 0.05) decreased in the skin tissues collected at 5 h (3.61 ± 1 and 4.2 ± 1, respectively) and 24 h postmortem (2 ± 0.7 and 2.4 ± 0.8, respectively) (Table 2 and Figure 4).

Multivariate analysis of variance was used to examine the associations between the post-wound infliction duration, postmortem interval and lncFAO gene expression. It showed a significant multivariate effect regarding the post-wound infliction duration and the lncFAO gene expression (0 day versus 8th and 10th day, 2nd day versus 8th and 10th day, and 4th day and 10th day: Wilks’ Lambda of 0.08, F = 5.7, and *p* < 0.001). In addition, postmortem interval had a significant effect on lncFAO gene expression (0 h versus 5 h and 24 h: Wilks’ Lambda of 0.1, F = 39.6, and *p* < 0.000). 

## 5. Discussion 

Loss of the cutaneous continuity due to trauma may lead to serious complications and even death [19]. Therefore, determination of the wound vitality and dating is a crucial medicolegal issue, which raises the need for evaluating the well-organized wound healing sequences and the involved biomarkers [20]. 

The progression of wound healing was confirmed in this study using morphological and histopathological methods where maximal inflammatory parameters, including edema, congestion and inflammatory infiltrate with re-epithelialization and remodeling initiation were found at 4 days post-wounding. However, cellular microscopic examination did not show autolytic changes in the studied early PMI (24 h). 

During wound healing, similar observations were reported [1,21,22]. Polymorphs migration was initiated at day 1, while monocytes were observed in the injured area at days 1, 3, and 8. The activated inflammatory cells secreted proinflammatory cytokines that attracted fibroblast into the wound at day 2 and enhanced fibroblasts migration at day 5 to produce extra cellular matrix that consisted mainly of collagen [23,24]. 

Collagen deposition was observed in our study at day 2 in the form of slightly stained meshwork, which became thick mature collagen at day 10. Early postmortem duration did not show collagen homogenizations affect.

After 2–3 days of wound healing, the attracted fibroblasts became involved in granulation tissue synthesis, allowing neovascular formation and leukocytes migration. Hyaluronic acid, elastin, proteoglycans and procollagen, which aggregated into ordered fibrillar structure, are the main granulation tissue’s components [25]. 

During wound healing, collagen type and amount are continuously changed to determine healed skin’s tensile strength. Collagen III is an early response collagen, which is replaced by collagen I, and then collagen maturation is enhanced to produce complex mature collagen that maintains cutaneous integrity and determines its tensile strength [23,24].

Appropriate wound healing needs a balance between new matrix synthesis and matrix metalloproteinase degradative activities [25].

In the current study, MMP-9 was increased till 8 days post-wounding, then it significantly reached its lower level at 10 days after wounding. 

The natural tissue concentrations of gelatinases, including MMP-9, are normally low, whereas their production is highly enhanced after injury. Their biological activity is upregulated by various factors, including tissue hypoxia, and inflammatory cytokines, such as tumor necrosis factor-α, and interleukin (IL) (IL-1 and IL-8). Nevertheless, its biological activity is downregulated by tissue inhibitor of metalloproteinase-1. [10,26]. 

MMP-9 is continuously produced by various cells, including tissues macrophages [11]. After wounding, MMP-9 is detected in injured tissue on the day of injury and decreased to allow advanced healing process [27]. 

Its early accumulation in the wound area is needed mainly for collagen/gelatin cleavage, pro-/antiangiogenic factors processing, keratinocytes migration, endothelial progenitor cells mobilization and vasculogenesis. However, its prolonged expression could be associated with impaired wound healing [11]. Reiss et al. noted that the excessive expression of MMP-9 in the wound after 12 days post-wounding significantly affected the rate of healing and might have prevented and limited epithelial migration leading to failure of wound closure [28]. Moreover, the dynamic balance between MMP-9 and its inhibitor factor is needed to prevent prolonged inflammation, secondary tissue damage, and delayed wound healing [29]. 

MMP-9 was detected in postmortem injured samples up to 24 h after death and its expressions were not distinguished from ante-mortem expressions. 

In a study on cardiac muscle, Niedecker et al. [8] detected MMP-9 expression in experimentally antemortem and postmortem wounds for up to 3 h. 

After death, cells are still alive during molecular life, and ATP levels influence the time of their death. This indicates the presence of the inflammatory markers and their postmortem reaction during early postmortem durations [30]. 

In the current study, the observed rate of microscopic and immunohistochemical postmortem autolytic alterations was slower than that observed by KHALAF et al. They observed that mild to moderate postmortem autolytic changes occurred in postmortem cutaneous wounds at day 0 and 1, respectively. However, Wei et al. did not observe any remarkable histopathological changes in human skin kept at 4–6 °C within early 24 h postmortem; in addition, they stated that cellular degeneration and nuclear fragmentation need a longer duration to appear.

The rate of postmortem changes is either increased or decreased by multiple internal factors, such as body mass index, surface area, infection, injury type and size. In addition, there are external factors, such as geographical conditions. Understanding these factors and their effects is vital in PMI estimation [4]. 

Even though morphological, histopathological and immunohistochemical changes in the skin with the passage of time after death are used to assess the PMI in medicolegal researches, they are not sufficient for early PMI estimation and can be used as a preliminary method only, which should be combined with other biomolecular parameters for more precise results [31].

Therefore, the current study was also conducted to evaluate the expressions of lncFAO, which were strongly associated with healing sequences, especially inflammation resolution and tissue repair [5,20,32]. We identified its specificity in wound dating estimation and stability during early PMIs (up to 24 h). For specificity, its expressions were identified immediately after wounding, which significantly increased to reach the highest concentrations at 10th day post-infliction; the remodeling stage.

lncFAO can be detected in low concentrations in many normal tissues, but its levels increases after injury. lncFAO RNA regulates inflammation resolution by macrophages that have an extended variety of roles in wound healing, including inflammation initiation, resolution, and tissue remodeling [5]. Dolfi et al. identified that the high fatty acid oxidation rate was exhibited in the activated macrophages that can be isolated from tissues after long PMIs (18 h) [33].

Nakayama et al. acknowledged the role of lncFAO in inflammation resolution by macrophage. They found that lncFAO deficient mice exhibited increased expression of inflammatory response genes, including IL1 and IL6. They also had down-regulated macrophage transition from pro-inflammatory (M1) phenotype to anti-inflammatory (M2) phenotype; mitochondrial function was impaired through dysregulation of hydroxyacyl-coenzyme A dehydrogenase trifunctional multienzyme complex subunit beta (HADHB) that catalyzed the final step of β-oxidation [5,18,34,35]. 

For stability, lncFAO was detected in postmortem specimens up to 24 h; however, its expression was significantly negatively correlated with PMI. The highest significant difference was mainly noticed between 0 h and 24 h PMI in each post-wounding group, which assumed that lncFAO expression could be used in the PMI determination within 24 h. 

According to Sun et al. [36], who assessed Fosl1 mRNA in antemortem (up to 48 h) and post mortem (up to 24 h) in muscular samples, mRNA expression was increased in antemortem injured samples and decreased in postmortem samples in a time-dependent manner.

The expression of different RNA is influenced by wound infliction and is suitable for assessing postmortem wound dating in early phases. The rate of postmortem RNA degradation is affected by various intrinsic and extrinsic factors. However, RNA, including miRNA and lncRNA, of sufficient quality and quantity were isolated from long preserved forensic samples [20,36]. 

Macrophages have different phenotypes with unique functions during wound healing and their transition from M1-phenotype to M2-phenotype can be easily achieved by different inflammatory cytokines, including IL-4, IL-10, or IL-13, and various pathways of signal transduction, transcriptional and post-transcriptional regulatory networks, such as miRNAS, e.g., miR-27a-5p, let-7c-1-3p, and lncRNAS, e.g., lncFAO, lnc XIST, and lnc GAS5 [15,37,38,39] (Figure 5).

During the inflammatory phase, the wound is filled with pro-inflammatory macrophages of phagocytose cellular debris cells and pathogens to prepare the wound for healing by producing proinflammatory cytokines and proteolytic enzymes, including metalloproteinases, especially MMP-2 and MMP-9 (Figure 5) [1,26].

During the proliferation phase, the macrophage populations begin transitioning to anti-inflammatory macrophages that secrete factors, including metalloproteinases, especially MMP-1, MMP-8, and MMP-9 that enhance vascular formation, granulation tissue formation, collagen deposition, and re-epithelialization (Figure 5) [10].

During the remodeling phase, the wound is filled with anti-inflammatory macrophages that enhance granulation tissue breakdown and allow collagen maturation and improve regenerated skin strength in which the down-regulation of a variety of MMPs, including MMP-9, and up-regulation of various lncRNAs, including lncFAO, are involved (Figure 5) [5,9,10,37]. 

Despite the presence of some limitations, such as using the animal model, the results of this initial study elaborate the possible use of lncFAO as a reliable indicator for postmortem interval estimation. However, further research is needed to explore this marker in human cutaneous samples from different body parts and of different ages to verify the results of this preliminary study. 

Although the main target of this study was to assess the effect of early postmortem interval on certain biomarkers in excisional wound model, there remains the possibility that other factors may modulate the postmortem effect which should be explored, including wound type, health status, post-wounding and postmortem durations, etc.

## 6. Conclusions

The postmortem evaluation of immuno-histopathological changes and lncFAO gene expression for incisional cutaneous wound vitality and dating determination indicated that inflammatory, proliferative, and re-modelling changes are similar to the antemortem wounds at different post-wounding time. The expression of long noncoding fatty acid oxidation RNA in injured cutaneous samples was found to be increased in a time-dependent manner during the healing process and was significantly decreased with PMI prolongation. Although certainly inconclusive, lncFAO appears to be useful for early PMI estimation in combination with other biomolecular markers.

## Figures and Tables

**Figure 1 biomedicines-10-02919-f001:**
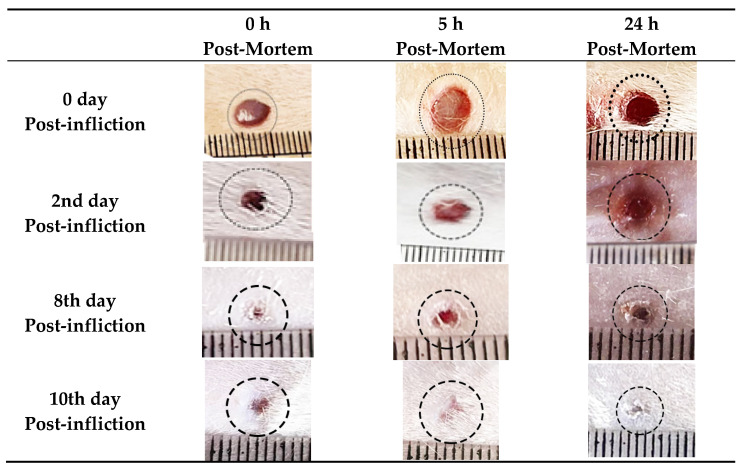
Photograph showing wound healing progression on day 0, day 2, day 4, day 8, and day 10 post-wounding at different postmortem intervals (0, 5th, and 24th h); each grade in the scale equal 1 mm.

**Figure 2 biomedicines-10-02919-f002:**
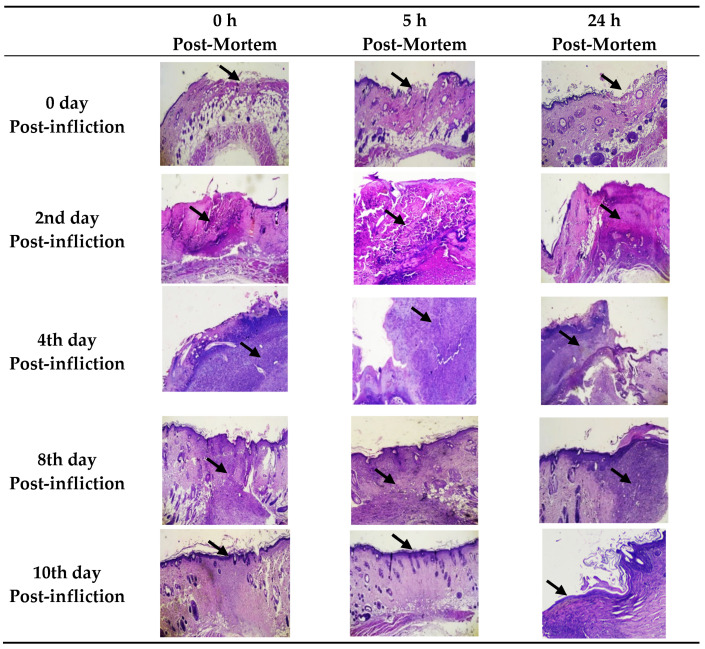
Photomicrograph of wound healing progression (40× magnification) in day 0, day 2, day 4, day 8, and day 10 post-wounding at different postmortem intervals (0, 5th, and 24th hours). Day 0 showed epidermal ulceration (arrow) with mild dermal edema, while on the 2nd day, there were moderate edema, congestion and inflammatory cellular infiltrate with mid fibroplasia (arrow). On the 4th day, wounds showed marked edema, congestion and inflammatory cellular infiltrate with moderate fibroplasia (arrow) and early attempts of re-epithelialization, and on the 8th day, it showed mild to moderate edema, congestion, and inflammatory cellular infiltrate with moderate fibroplasia (arrow) and near complete re-epithelialization. On the 10th day, wounds showed complete re-epithelialization (arrow) with disappearance of inflammatory parameters. No obvious change was detected at different early post-mortem intervals.

**Figure 3 biomedicines-10-02919-f003:**
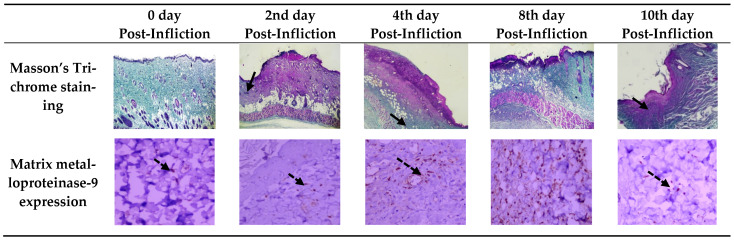
Representative of Masson Trichome staining (magnification, ×40) and metalloproteinase-9 expression (magnification, ×400) of cutaneous wound lesions on 0, 2nd, 4th, 8th, and 10th days post-wounding where wound at 0 day showed. Collagen was represented with black arrows, and matrix metalloproteinase-9 was represented with dashed black arrows.

**Figure 4 biomedicines-10-02919-f004:**
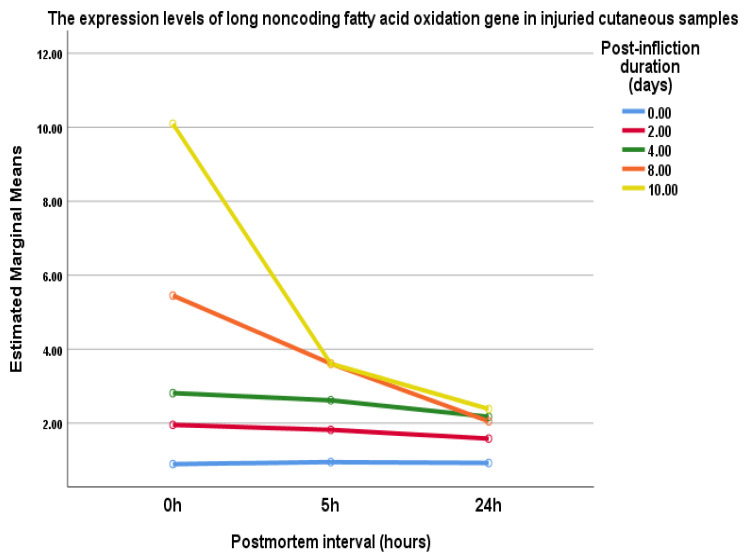
The expression of lncFAO gene in 0, 2, 4, 8, 10 days after wound infliction at different postmortem intervals (0, 5th, and 24th h).

**Figure 5 biomedicines-10-02919-f005:**
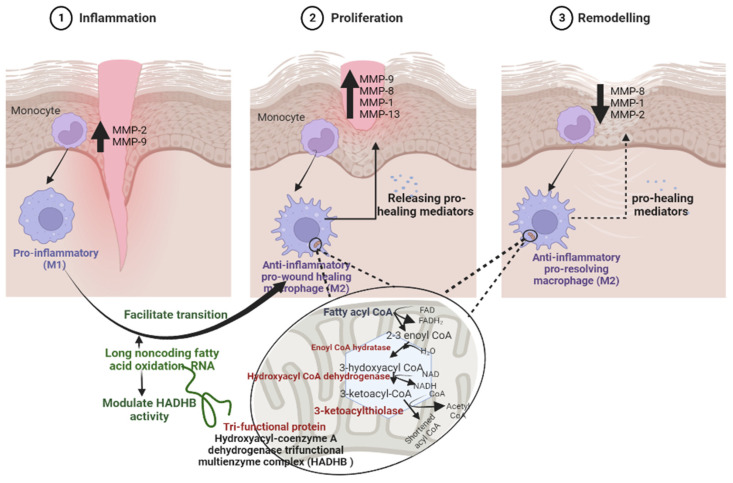
A schematic presentation of macrophage driven acute wound healing. During its three phases; inflammation, proliferation and remodeling. Pro-inflammatory macrophages phagocytose cellular debris and pathogens and release proinflammatory cytokines along with proteolytic enzymes, including metalloproteinases, especially MMP-2, and MMP-9. The macrophage populations begin transitioning to anti-inflammatory pro-wound healing macrophages; this transition is up-regulated by lncFAO that secrete factors, including metalloproteinases, especially MMP-1, MMP-8, MMP-9, that enhance vascular formation, granulation tissue formation, collagen deposition, and re-epithelialization. The anti-inflammatory pro-resolving macrophages enhance granulation tissue breakdown, allow collagen maturation, and improve regenerated skin strength along with down-regulation of a variety of MMPs, including MMP-9. Below the diagram, the main mechanism of lncFAO during the macrophage transition process through activating the β-subunit of mitochondrial trifunctional protein (hydroxyacyl-CoA dehydrogenase/3-ketoacyl-CoA thiolase/enoyl-CoA hydratase β-subunit; HADHB), a keyenzyme in fatty acid oxidation.

**Table 1 biomedicines-10-02919-t001:** Scoring of histopathological findings of injured cutaneous samples.

Post-Wounding Time	Congestion	Edema	Hemorrhage	Cellular Infiltration	Fibroblast Proliferation	Angiogenesis	Epithelization	Collagen
**0 day**	-	+	-	-	-	-	-	-
**2nd day**	++	++	+	++	+	-	-	-
**4th day**	+++	+++	++	+++	++	+	+/++	-
**8th day**	++	++	+	+/++	++	++	++/+++	++
**10th day**	-	-	-	-	-	-	+++	+++

No alteration (-), mild alterations (+), moderate alterations (++), maximal alterations (+++), mild to moderate alterations (+/++), moderate to maximal alterations (++/+++).

**Table 2 biomedicines-10-02919-t002:** Long noncoding fatty acid oxidation expression (mean ± SD) in the examined cutaneous samples at different post-wounding time interval and postmortem intervals.

	Postmortem	0 h	5 h	24 h
Post-Wounding				
**0 day**		0.9 ± 0.2	0.95 ± 0.3	0.92 ± 0.3
**2nd day**		1.95 ± 0.7 *	1.8 ± 0.8 *	1.6 ± 0.5 *^,#^
**4th day**		2.8 ± 0.8 *	2.6 ± 0.3 *	2.2 ± 0.2 *^,#^
**8th day**		5.45 ± 0.5 *	3.61 ± 1 *^,#^	2 ± 0.7 *^,#^
**10th day**		10 ± 2.5 *	4.2 ± 1 *^,#^	2.4 ± 0.8 *^,#^

Data presented as mean ± SE, * Statistically significant compared to corresponding value in 0 day (*p* < 0.05). ^#^ Statistically significant compared to corresponding value in 0 h (*p* < 0.05).

## Data Availability

Upon request, the data available within article are obtainable from the corresponding authors.

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
