# Peer review of "Estimation of Early Postmortem Interval from Long Noncoding RNA Gene Expression in the Incised Cutaneous Wound: An Experimental Study"

_biomedicines, 2022, doi:10.3390/biomedicines10112919_

Round 1

Reviewer 1 Report

ag 1 lines 39-40: on which ground authors assert that PMI estimation is more important in cases involving several inflicted injuries? In my opinion post-mortem interval estimation is paramount in all forensic scenarios;

Pag 1 and 2 lines 46-49: in the introduction authors report that the tools they want to validate in this paper have been previously established to have a time related behavior. If so, what is new?

Pag 3 line 103: I suggest authors to explain in more detailed way the process used to arbitrarily assign the wound-score. It’s not clear on what magnification and how many fields of the specimen were used to make the score homogenous and comparable? 10x magnification and 10 fields? 40X magnification and 5 fields? If not stated, the results may not be reproduced in other labs and even the statement of a ‘semi-quantitative’ value of the score may vanish.

Pag 7 figure 4: this histogram is very difficult to understand, due to the fact that the variable reported in the x-axis are 4 while in the caption there are reported only 3 time points (0, 5 and 24 hours)

Pag 3 line 112: as for histopathological examination, even the MMP-9 expression may result very difficult to reproduce.

Pag 8 lines 283-287 and 292-296: The two consecutive sentences are in contrast one to the other. Which is the biological truth? MMP-9 increases in the first 24 or in 192 hours?

Pag 9 lines 315-318: the affirmation is untrue. Neither morphological nor histopathological and/or immunohistochemical changes have been so far used to thanato-chrono-diagnosis.

Pag 9 lines 325-330: redundant.

Pag 9 lines 336-339: this affirmation is unproven, the data shown do not confirm this assumption.

Author Response

I appreciate your effort and valuable comments to improve the quality of the manuscript

ag 1 lines 39-40: on which ground authors assert that PMI estimation is more important in cases involving several inflicted injuries? In my opinion post-mortem interval estimation is paramount in all forensic scenarios;

Reply: I agree with your suggestion and it was corrected as advised

Pag 1 and 2 lines 46-49: in the introduction authors report that the tools they want to validate in this paper have been previously established to have a time related behavior. If so, what is new?

Reply: We discussed the role of lncFAO to be used as a marker in detecting time passed since death in early postmortem interval 24hs. 

Pag 3 line 103: I suggest authors to explain in more detailed way the process used to arbitrarily assign the wound-score. It’s not clear on what magnification and how many fields of the specimen were used to make the score homogenous and comparable? 10x magnification and 10 fields? 40X magnification and 5 fields? If not stated, the results may not be reproduced in other labs and even the statement of a ‘semi-quantitative’ value of the score may vanish.

Reply: thanks for the comment. this statement was added “each parameter was evaluated at 40x magnification in 5 fields and was categorized from 0 to 3 that ranged from no to marked alterations respectively.”

Pag 7 figure 4: this histogram is very difficult to understand, due to the fact that the variable reported in the x-axis are 4 while in the caption there are reported only 3 time points (0, 5 and 24 hours)

Reply: I noticed the point and the histogram was changed as advised to be more clear

Pag 3 line 112: as for histopathological examination, even the MMP-9 expression may result very difficult to reproduce.

Reply: before starting the research, we do a literature review and the histopathological examination and evaluation were already done according to Khalaf et al., 2019 and Abu-Albasal et al, and MMP-9 expression evaluation was alreadu done according to Niedecker et al., 2021. the method was already reproduced in our lab so it can be followed in the future. 

  1. Khalaf AA, Hassanen EI, Zaki AR, Tohamy AF, Ibrahim MA. Histopathological, immunohistochemical, and molecular studies for determination of wound age and vitality in rats. International wound journal. 2019;16(6):1416-25.
  2. Abu-Al-Basal MA. Healing potential of Rosmarinus officinalis L. on full-thickness excision cutaneous wounds in alloxan-induced-diabetic BALB/c mice. Journal of ethnopharmacology. 2010;131(2):443-50.
  3. Nakayama Y, Fujiu K, Yuki R, Oishi Y, Morioka MS, Isagawa T, et al. A long noncoding RNA regulates inflammation resolution by mouse macrophages through fatty acid oxidation activation. Proceedings of the National Academy of Sciences. 2020;117(25):14365-75.

Pag 8 lines 283-287 and 292-296: The two consecutive sentences are in contrast one to the other. Which is the biological truth? MMP-9 increases in the first 24 or in 192 hours?

Reply: Two statements were revised and rewritten to be more clear

Pag 9 lines 315-318: the affirmation is untrue. Neither morphological nor histopathological and/or immunohistochemical changes have been so far used to thanato-chrono-diagnosis.

Reply: I appreciate your remark. It is corrected to be “ in medicolegal researches

Pag 9 lines 325-330: redundant.

Reply: It is re-written

Pag 9 lines 336-339: this affirmation is unproven, the data shown do not confirm this assumption.

Reply: I appreciate your remark. It is revised

Reviewer 2 Report

I find the idea for the work very interesting; but there are some major limitations; different accident sequences have a decisive impact on wound healing and also on the corresponding biomarkers. Burns lead to different changes than cuts,.... Age also has an influence on wound healing and also on the biomarkers, as do some diseases, etc. These and other limitations are not really mentioned and discussed. Numerous studies are needed.

Author Response

I find the idea for the work very interesting; but there are some major limitations; different accident sequences have a decisive impact on wound healing and also on the corresponding biomarkers.

Reply: thanks for your support

Burns lead to different changes than cuts,.... Age also has an influence on wound healing and also on the biomarkers, as do some diseases, etc. These and other limitations are not really mentioned and discussed.

Reply: thanks for the valuable comment and It is added in the limitation section

Numerous studies are needed.

Reply: Discussion was reviewed and more studies are added

English proof-reading was done

Reviewer 3 Report

The present work examines the progression of wound healing in rats using morphological and histopathological methods. The expression of long noncoding RNAs expression such as lncFAO RNA), matrix metalloprotease-9 was studied during different early postmortem intervals. The experimental methods in this manuscript could be better discussed based on the results. I think the following revisions are necessary.

1.       Introduction is not sufficient either for MMPs or long noncoding RNAs and should be strengthened.

2.       The morphological and histopathological assessment of the wound in different time periods

3.       The authors checked only one MMP (MMP-9). What about other members or the group;

4.       The authors could evaluate the expression of inflammatory markers during the post-mortem interval.

5.       The English grammar should be carefully checked. Also, overall the draft particularly within the discussion section. Indeed, the Results section requires much additional work in terms of structure and writing quality. 

6.       The author could add an abbreviation section.

Author Response

The present work examines the progression of wound healing in rats using morphological and histopathological methods. The expression of long noncoding RNAs expression such as lncFAO RNA), matrix metalloprotease-9 was studied during different early postmortem intervals. 

Reply: thanks for support and suggestion for manuscript improvement 

The experimental methods in this manuscript could be better discussed based on the results. I think the following revisions are necessary.

  • Introduction is not sufficient either for MMPs or long noncoding RNAs and should be strengthened.

Reply: the introduction section was revised as advised

  • The morphological and histopathological assessment of the wound in different time periods

Reply: before starting the research conduction, a literature review was done and our method followed Khalaf et al., 2019, Abu-Albasal et al, and Niedecker et al., 202.

  1. Khalaf AA, Hassanen EI, Zaki AR, Tohamy AF, Ibrahim MA. Histopathological, immunohistochemical, and molecular studies for determination of wound age and vitality in rats. International wound journal. 2019;16(6):1416-25.
  2. Abu-Al-Basal MA. Healing potential of Rosmarinus officinalis L. on full-thickness excision cutaneous wounds in alloxan-induced-diabetic BALB/c mice. Journal of ethnopharmacology. 2010;131(2):443-50.
  3. Nakayama Y, Fujiu K, Yuki R, Oishi Y, Morioka MS, Isagawa T, et al. A long noncoding RNA regulates inflammation resolution by mouse macrophages through fatty acid oxidation activation. Proceedings of the National Academy of Sciences. 2020;117(25):14365-75.
  • The authors checked only one MMP (MMP-9). What about other members or the group.

Reply: I appreciate your comment and the some role of other members were added in the introduction and during the discussion section. checking them will be a good study point that can be included in future researches

  • The authors could evaluate the expression of inflammatory markers during the post-mortem interval.

Reply: Reply: I appreciate your comment and the some role of inflammatory markers were added in the introduction and during the discussion section. checking them will be a good study point that can be included in future researches

  • The English grammar should be carefully checked. Also, overall the draft particularly within the discussion section. Indeed, the Results section requires much additional work in terms of structure and writing quality. 

Reply: English proof-reading was done and manuscript was revised

  • The author could add an abbreviation section.

It was added as suggested

Reviewer 4 Report

My suggestions:

1. How old were the rats at the time of the experiment?

2. Besides MMP9 are there any other MMPs impact wound healing? It would be nice to explain briefly in the introduction. 

3. In the discussion I would add a potential pathway figure, which may explain, how MMP9 and lncFAO could impact wound healing. 

4. In the discussion and conclusion I would explain a little bit more in detail, how MMP and LncFAO could be related to each other. Are they sharing some kind of common mechanisms? 

5. Were there any other MMPs and long non-coding RNAs analyzed in case of wound healing?

Author Response

Thanks for your support and valuable comments for improving manuscript quality

  1. How old were the rats at the time of the experiment?

Reply: 8 weeks

2. Besides MMP9 are there any other MMPs impact wound healing? It would be nice to explain briefly in the introduction. 

Reply: introduction was revised as advised

3. In the discussion I would add a potential pathway figure, which may explain, how MMP9 and lncFAO could impact wound healing. 

Reply: it was added

4. In the discussion and conclusion I would explain a little bit more in detail, how MMP and LncFAO could be related to each other. Are they sharing some kind of common mechanisms? 

Reply: They are indirectly related and connected via macrophage transition and activation. This is highlighted in the discussion section

5. Were there any other MMPs and long non-coding RNAs analyzed in case of wound healing?

Reply: Yes they were added in the introduction and discussion sections

Round 2

Reviewer 2 Report

No further comments

Reviewer 4 Report

Authors fulfilled my suggestions, thank you.